



# Three-in-one: GPS-IR measurements of ground surface elevation changes, soil moisture, and snow depth at a permafrost site in the northeastern Qinghai-Tibet Plateau

Jiahua Zhang[1], Lin Liu[1], Lei Su[2,3], Tao Che[2,4]

[1]Earth System Science Programme, Faculty of Science, The Chinese University of Hong Kong, Hong Kong, 999077, China
[2]Heihe Remote Sensing Experimental Research Station, Key Laboratory of Remote Sensing of Gansu Province, Northwest Institute of Eco-Environment and Resources, Chinese Academy of Sciences, Lanzhou, 730000, China
[3]University of Chinese Academy of Sciences, Beijing, 100049, China
[4]Center for Excellence in Tibetan Plateau Earth Sciences, Chinese Academy of Sciences, Beijing, 100101, China

*Correspondence to*: Tao Che (chetao@lzb.ac.cn)

**Abstract.** Ground surface elevation changes, soil moisture, and snow depth are all essential variables for studying the dynamics of the active layer and permafrost. GPS interferometric reflectometry (GPS-IR) has been used to measure surface elevation changes and snow depth in permafrost areas. However, its applicability to estimating soil moisture in permafrost

regions has not been assessed. Moreover, these variables were usually measured separately at different sites. Integrating their estimates at one site facilitates the comprehensive utilization of GPS-IR in permafrost studies. In this study, we run simulations to elucidate that the commonly-used GPS-IR method for estimating soil moisture content cannot be directly used in permafrost areas, because it does not consider the bias introduced by the seasonal surface elevation changes due to thawing of the active layer. We propose a solution to improve this default method by introducing modeled surface elevation

changes. We validate this modified method using the GPS data and in situ observations at a permafrost site in the northeastern Qinghai-Tibet Plateau (QTP). The root-mean-square error and correlation coefficient between the GPS-IR estimates of soil moisture content and the in situ ones improve from 1.85% to 1.51% and 0.71 to 0.82, respectively. We also implement a framework to integrate the GPS-IR estimates of these three variables at one site and illustrate it using the same site in the QTP as an example. This study highlights the improvement to the default method, which makes the GPS-IR valid

in estimating soil moisture content in permafrost areas. The three-in-one framework is able to fully utilize the GPS-IR in permafrost areas and can be extended to other sites such as those in the Arctic. This study is also the first to use GPS-IR to

estimate environmental variables in the QTP, which fills a spatial gap and provides complementary measurements to those of ground temperature and active layer thickness.

## 1 Introduction

Permafrost refers to the ground where the temperature remains at or below 0 °C for at least two consecutive years. On top of the permafrost is an active layer which undergoes seasonal freezing/thawing cycles (Dobinski, 2011). In the Qinghai-Tibet Plateau (QTP), permafrost occupies around 40% of its area (Zou et al., 2017) and has been warming and degrading over the last several decades (Zhao et al., 2010 and 2019). The ground temperature at 15 m depth increased at a rate varying from 0.02 °C per decade in Hoh Xil to 0.26 °C per decade in the Kunlun Mountains and Liangdaohe during 2001–2017 (Zhao et

al., 2019). Thickening of the active layer was also observed. Based on the records at 10 sites, the average thickening rate was 19.5 cm per decade from 1981 to 2018 (Zhao et al., 2019). The dynamics of the active layer and permafrost (collectively called as frozen ground alternatively) has a crucial impact on geomorphological, hydrological, ecological processes and infrastructures (Wu et al., 2002; Cheng and Wu, 2007; Yang et al., 2010; Gao et al., 2017).

Ground surface elevation changes, soil moisture, and snow depth are all essential variables for studying the dynamics of the frozen ground, as they are all related to the thermal and hydrological changes in the frozen ground. The ground surface in permafrost areas is subject to uplift/subsidence, mainly due to the phase changes between ice and water in the active layer freezing/thawing cycles. Surface deformation can indicate the variations in the active layer and permafrost. Numerous studies have been conducted to use surface elevation changes (e.g., Interferometric Synthetic Aperture Radar (InSAR)

measurements) to infer the variation of active layer thickness and permafrost degradation in the QTP (Chen et al., 2018; Wang et al., 2018; Reinosch et al., 2020; Daout et al., 2020). Soil moisture affects soil thermal properties then the ground thermal regime, for instance, in well-drained areas with peat layers, decreasing soil moisture content lowers the soil thermal conductivity, as a large amount of pore space is filled with air. Such a layer functions as an insulator, retarding the heat transfer between the atmosphere and the lower ground and thereby impeding the downward movement of the thawing front

(Shiklomanov et al., 2010; Göckede et al., 2019). Surface soil moisture also regulates the exchange of water and energy

between the atmosphere and frozen ground by evapotranspiration (Seneviratne et al., 2010). Zhang et al. (2016) investigated

the influence of soil moisture on the thermal and hydrological properties of the active layer in Tanggula in the central region

of the QTP. Snow cover also has a significant influence on the ground thermal regime due to its high albedo, low thermal

conductivity, and the large amount of latent heat absorbed when melting (Zhang, 2005). The timing and duration of snow

cover, snow depth, and snow density and texture affect the frozen ground dynamics. The variations of snow cover and their

impact on the thermal and hydrological conditions of the frozen ground in the QTP have been studied extensively (Flanner

and Zender, 2005; Che et al., 2008; Gao et al., 2012; Xu et al., 2017; Deng et al., 2017, Qi et al., 2019).

GPS interferometric reflectometry (GPS-IR) is a method which exploits the interference between direct and reflected GPS

signals to estimate ground surface elevation changes, surface soil moisture, and snow depth (Larson, 2019). Such

interference can be reflected by signal-to-noise ratio (SNR) data at low satellite elevation angles recorded by GPS receivers.

SNR observations (hereafter called SNR interferograms) oscillate quasi-sinusoidally when the reflecting surface is relatively

smooth and horizontal. The frequency of such oscillation (hereafter referred to simply as "frequency") can be used to

measure the distance between the antenna and reflecting surface, which is then converted into snow depth when the reflector

is  snow surface while the ground surface elevation changes (Larson et al., 2009 and 2016; McCreight et al., 2014; Liu and

Larson, 2018). The phase of the oscillation (hereafter referred to simply as "phase") can be used to estimate the surface soil

moisture content within the layer of 0–5 cm (Larson et al., 2008; Chew et al., 2014 and 2016). GPS-IR can provide daily and

continuous measurements at continuously operating sites. Their spatial coverages are antenna-height dependent, e.g., ~1000

$m^2$ for a 2m-high antenna. Such an order of magnitude makes the GPS-IR measurements bridge the point observations and

regional-scale remote sensing ones.

The applicability of GPS-IR for estimating soil moisture content in permafrost areas has not been assessed. We run

simulations to elucidate that it could not be applied directly to permafrost areas as it does not consider the bias introduced by

the seasonal surface elevation changes due to the thawing of the active layer. Moreover, measuring ground surface elevation

changes, soil moisture content, and snow depth at a single site can fully utilize GPS-IR in permafrost areas and boost the

permafrost studies, which however has not been conducted to date. Driven by these motivations, our objectives in this study are (1) to improve the default GPS-IR method for estimating soil moisture content to make it valid in permafrost areas; (2) to implement a three-in-one framework, i.e., integrating the GPS-IR measurements of surface elevation changes, soil moisture, and snow depth at one site, and illustrating it using a permafrost site within the northeastern QTP site as an example and (3)

to provide GPS-IR measurements at the QTP site.

The significance of this study relies on the improvement made to the default method, which can correct the bias introduced by the seasonal surface deformation. In summer, as the thawing front advances downwards, surface subsidence accumulates and leads to bias with larger magnitude. The bias would likely mislead the interpretation of soil moisture variation by

superimposing a seasonal trend. The modified method is also the basis of the three-in-one framework, which helps to fully utilize GPS-IR in permafrost areas and can be extended to other sites. Moreover, this study is the first to use GPS-IR for estimating environmental variables in the QTP. Although numerous GPS stations (e.g., the stations of the Crustal Movement Observation Network of China, http://www.neiscn.org/chinzdinfo/jsp/main.jsp) are continuously operating in the QTP, none of them have been previously used for GPS-IR studies. Furthermore, permafrost is extensive in the QTP, but the monitoring

sites (e.g., boreholes) are few and unevenly distributed (Zou et al., 2017). Our study site can fill a spatial gap in the QTP, although it was initially designed to study hydrological processes. In addition, our GPS-IR measurements are complementary to the existing observations, such as ground temperature, to provide fresh insights into frozen ground dynamics.

In section 2, we briefly describe the study site and instrumentation. In section 3, we first summarize the GPS-IR principles

for retrieving ground surface elevation changes, soil moisture, and snow depth. We then illustrate the default method's limitations for estimating soil moisture in permafrost areas by simulations and introduce our solution for improvements. We then propose a three-in-one framework. We finally present the datasets used in this study. In section 4, we show the results, i.e., the improvements in our method and the GPS-IR estimates of these three variables at the site in the QTP. In section 5, we discuss the merits and the possible error sources of the modified method and the benefits of the three-in-one framework

to permafrost studies. We conclude this study in section 6.

## 2 Description of the study site

The study site, Binggou (38.01°N, 100.24°E, 4120 m a.s.l.) is located in the northeastern region of the QTP (Fig. 1(a)). The mean annual ground surface temperature is around -3.5°C. Permafrost is present at this site with a thickness of 25–30 m (Ran et al., 2018). The active layer thickness is ~1.6 m, based on ground temperature observations. The biome at this site is alpine

steppe. Regarding the soil texture profile, in the upper 0.2 m, the soil is dominantly sandy silt. At depths between 0.2 m and 0.5 m, it is a mixture of sand, silt, and gravel. And the soil becomes mainly gravel at the depth of 0.5 m. This general soil texture description was kept when installing the GPS monument. The soil moisture content is ~40% in the upper 0.4 m, decreases to 20% at the depth of 0.8 m, and remains relatively stable to the depth of 1.6 m (Che et al., 2019).

A GPS station, called QLBG, has been operating in Binggou since November 2016. The antenna height is ~2 m above the ground surface, and the monument foundation is ~1.5 m deep. The foundation depth is slightly shallower than the active layer base. It implies that the monument might settle in late summer when the soil around the foundation starts to thaw (or heave up at the beginning of freezing season when the foundation freezes). However, the magnitude of such movement is expected to be negligible, as the layer between the foundation depth and the active layer base is as thin as ~10 cm. Given that

the soil moisture content is ~20%, the thawing of this layer causes a subsidence of only ~0.2 cm. Such a magnitude is at least one order less than the uncertainty of GPS-IR measurements. The monument can be regarded as stable in the thawing season considered in this study (see the detailed discussion in the supplementary). The antenna of QLBG is mounted onto a galvanized steel pipe anchored to a concrete foundation. The GPS receiver type is CHC N72, and the antenna is CHCC220GR with a CHCD radome. Figure 1(b) shows a ground photo of QLBG. An integrated weather station exists close

to QLBG, which records various environmental variables, including soil moisture and ground temperature (Che et al., 2019). They are both measured up to a depth of 1.6 m. Due to the open and relatively horizontal and smooth surface and abundant weather records, QLBG is usable for GPS-IR studies.



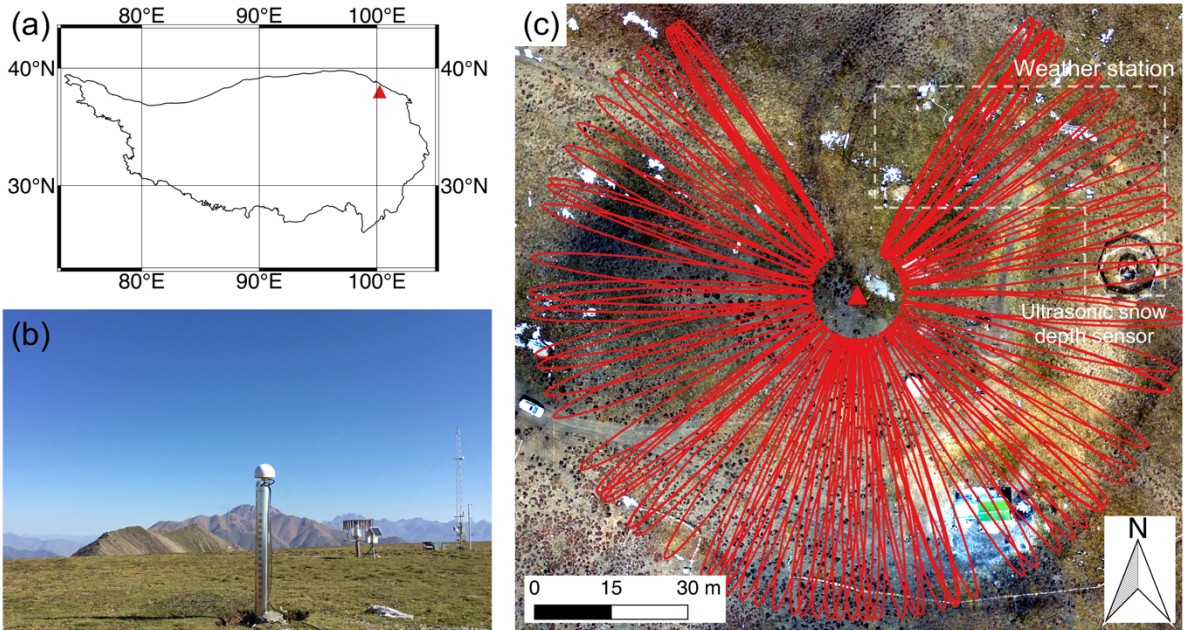

Figure 1: (a) Location of the study site, Binggou, in the northeastern Qinghai-Tibet Plateau. (b) Ground photo of the GPS station QLBG. (c) Orthophoto of the Binggou site showing the surface condition, QLBG (red triangle), and the integrated weather station. Red elliptic curves indicate the footprints of the reflected L1 GPS signals at the satellite elevation angle of 5°.

## 3 Methodology

### 3.1 GPS-IR

The input of GPS-IR is SNR data, which can reflect the interference pattern between the direct and reflected signals at low satellite elevation angles. When a GPS station is located above a horizontal and smooth surface (e.g., Fig. 2), the SNR interferogram, corresponding to a rising/setting satellite track, can be simply expressed as (Larson, 2019):

$$SNR = A(e)\sin\bigl(2\pi f \sin e + \phi(e)\bigr), \tag{1}$$

$$f = \frac{2H}{\lambda}, \tag{2}$$

where, $A(e)$ is the oscillation amplitude varying with satellite elevation angle $e$; $f$ is the oscillation frequency of the SNR interferogram; $H$ is the vertical distance between the antenna and the reflecting surface, conventionally called as reflector height; $\lambda$ is the carrier wavelength of GPS signals; $\phi(e)$ is the phase varying with satellite elevation angle as well. The





frequencies of SNR interferograms are used to obtain reflector heights by equation (2) then surface elevation changes in the snow-free season and snow depth (Larson et al., 2009; Liu and Larson, 2018). In addition, the phases are used to estimate

soil moisture content (Larson et al., 2008).

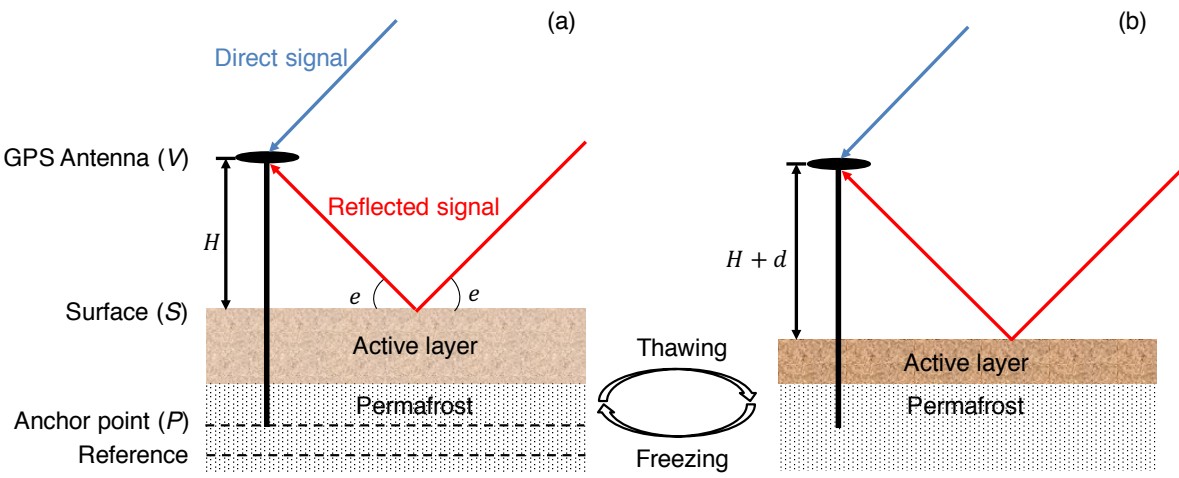

**Figure 2: Diagrams showing the geometries of GPS-IR in the thawing/freezing active layer conditions. We use the symbols $V$, $S$, and $P$ to denote the vertical positions of the GPS antenna, reflecting surface, and the monument anchor point with respect to an**

**arbitrary reference in the deep permafrost, respectively. $H$ denotes the reflector height. $d$ represents the surface deformation due to freezing/thawing of the active layer.**

In the geometry of GPS-IR (Fig. 2), reflector height $H$ depends on the vertical positions of the GPS antenna ($V$) and the reflecting surface ($S$) with respect to the deep permafrost as a reference, which is in the form of

$H = V - S$ ,                                                                                          (3)

Given the monument with a length of $L$, the position of the monument anchor point is

$P = V - L$ ,                                                                                          (4)

Incorporating (4) into (3), we can derive the surface position as

$S = L + P - H$ ,                                                                                      (5)





If the monument is stable with respect to the deep permafrost, the variation of $P$ relative to the reference is zero and the equation (5) will be reduced to

$$S = L - H ,  \qquad\qquad (6)$$

Assuming a constant $L$, surface elevation changes are negative to the reflector height variation.

In practice, to obtain reflector height, for any given SNR interferogram at low elevation angles (e.g., 5–15° used in this study), we first remove its 2nd-order polynomial fit and use the residuals, which are mainly contributed from the interference between direct and reflected signals. We then conduct Lomb-Scargle Periodogram on the residuals to obtain the frequency spectrum. We use the peak value of this spectrum to represent $f$ and convert it to $H$ by equation (2). On any given day, we retrieve $H$ from all available SNR interferograms. Then, we average them to obtain the daily reflector height $\bar{H}$ and use the
mean value's standard deviation to represent its uncertainty. When the ground is covered with snow, we convert the reflector height measurements into the snow depth, using that of the ground surface as the reference. In the snow-free season, based on equation (6), we remove the mean from the minus daily reflector heights and use the residuals to represent ground surface elevation changes.

Here, we describe the default method of estimating phases and then retrieving soil moisture content by GPS-IR. Practically, $A(e)$ and $\phi(e)$ of an SNR interferogram are assumed to be constant, as their variations with the elevation angle are small (Zavorotny et al., 2010; Chew et al., 2014). Fixing $H$ to an *a priori* reflector height ($H_0$), we can determine the phase by Least Squares Estimation (LSE). In previous studies such as Larson et al. (2010) and Chew et al. (2014), $H_0$ is the mean value of the daily reflector heights over the data time span of interest. On any given day within the time span, the same $H_0$ is
used to estimate phases of SNR interferograms. After retrieving the phases, we offset the phase time series of different satellite tracks by subtracting the mean value of the lowest 15%. Then, we use the mean values of the offset phases to represent the daily ones. The uncertainties are the standard deviations of the mean values. Then, we can convert the phases into soil moisture content, based on their empirical linear relationship (Chew et al., 2016; Small et al., 2016). For cases with no significant vegetation influence, the slope of soil moisture content versus phase is 1.48 cm$^3$ cm$^{-3}$ deg$^{-1}$ (Chew et al., 2014).



The intercept (or called as residual soil moisture content in literature) can be determined by in situ measurements or public

datasets (e.g., US Geological Survey's STATSGO (Schwarz and Alexander, 1995)).

### 3.2 Modifying the default method for estimating soil moisture in permafrost areas

### 3.2.1 Limitation of the default method

Using a constant $H_0$ to estimate phases in permafrost areas might not be valid, as the ground surface is subject to moves

vertically due to the active layer thawing/freezing. To illustrate this limitation and its impact on the phase estimation, we run

simulations by using the multipath simulator (Nievinski and Larson, 2014). We initially set the reflector height $H$ as 2 m,

which is the typical monument height of most GPS stations. Then, we introduce a surface deformation $d$. Positive (negative)

means surface subsidence (uplift). In the simulations, $d$ varies from -5 to 5 cm at a step of 1 cm. Not knowing the antenna

gain pattern of CHCC220GR CHCD used in this study, we alternatively use the one of TRM29659.00 with the radome of

SCIT. Other key parameters used in the simulations are listed in Table 1.

In Fig 3(a), we show an example of the simulated SNR with $d$ of 2 cm whose 3-order polynomial fit has been removed. We

use LSE to estimate the phase and amplitude by using $H$ and $H + d$, respectively. From the inset plot of Fig. 3(a), we can

observe that using $H$ introduces a phase bias of around -14°. Figure 3(b) shows the simulated bias corresponding to various

$d$. The bias is approximately proportional to the surface deformation.

In summer, surface subsidence accumulates with the downward movement of the thawing front, which leads to bias with

larger magnitude. The bias may mislead the interpretation of soil moisture variation, with a seasonal trend superimposed on

the soil moisture estimates. Thus, such bias needs to be corrected when using GPS-IR to estimate soil moisture content in

permafrost areas. To solve this problem, we propose a solution in the following subsection 3.2.2.

**Table 1: Key parameters for SNR simulations**

| Parameter | Value |
| --- | --- |
|  |  |





| GPS signal | L1 C/A |
| :---: | :---: |
| Antenna | TRM29659.00 with radome SCIT |
| Reflector height | 1.95–2.05 m with 1 cm intervals |
| Elevation angle | 5–20° |
| Azimuth angle | 0–360° |
| Soil type | Sandy loam |

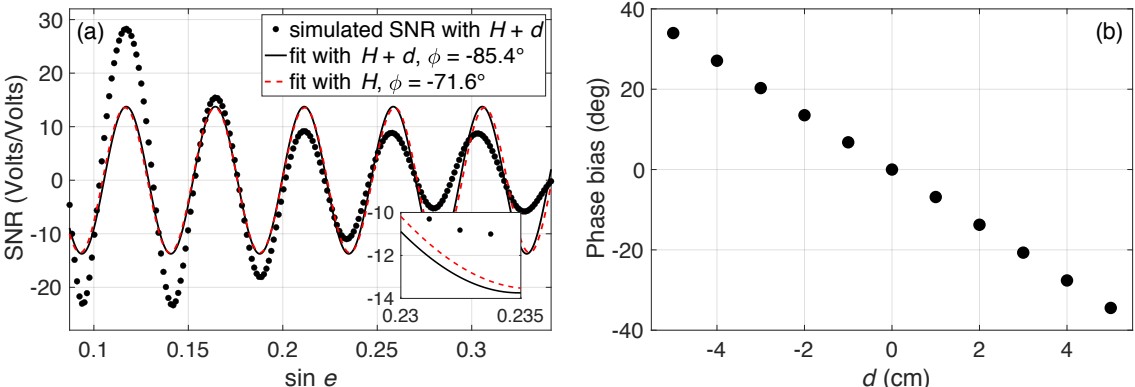

**Figure 3: (a) Simulated SNR with $H + d$ and their fits based on $H + d$ and $H$, respectively. In this simulation, $H$ and $d$ are fixed as 2 m and 2 cm, respectively. The inset plot shows that the phases of these two fits are different. (b) The simulated phase bias when using various $d$.**

### 3.2.2 Solution: introducing modeled ground surface elevation changes

To address the problem illustrated above, we propose a solution of introducing modeled ground surface elevation changes.

We add the modeled values to the constant $H_0$ to derive the time-varying reflector height $H_0'$, which approximates the true daily changes of reflector height. For any given day, we use its corresponding reflector height to estimate phases. We need to note that the daily reflector heights derived by the Lomb-Scargle Periodogram cannot be used directly. They cannot reflect the daily changes of reflector height, as their uncertainties are on the order of a few centimeters.





We simulate the ground surface elevation changes based on the Stefan-equation-based model of Liu et al. (2012). It can

estimate surface deformation on any given day from the onset of thawing season by using the GPS-IR-measured ground

surface deformation and thermal indices. The model is expressed as

$$d(t) = d_s \widetilde{I_T}(t) + d_0 , \tag{7}$$

$$I_T(t) = \sqrt{A_T(t)} , \tag{8}$$

$$A_T(t) = \sum T(t) \; when \; T(t) > 0 \, °C , \tag{9}$$

where $d(t)$ is the surface deformation on the day of $t$. $d_s$ is the seasonal subsidence. $\widetilde{I_T}(t)$ is the normalized thermal index

$I_T(t)$ by its maximum value. $I_T(t)$ is the square root of degree days of thawing, $A_T(t)$, from the onset of thawing season,

calculated based on the ground surface temperature $T$.

We use the GPS-IR-measured ground surface elevation changes and normalized thermal indices to inverse the parameters of

$d_s$ and $d_0$. They are then used together with the normalized thermal indices to simulate the ground surface elevation changes

and then obtain $H_0'$. We then use the $H_0'$ to estimate the phases. As in situ soil moisture observations are available, we

directly compare them to the phases to obtain the mapping function, which is used in turn to convert the phases to soil

moisture content.

**3.3 Framework of integrating GPS-IR measurements of ground surface elevation changes, soil moisture, and snow depth**

We propose a 3-in-1 framework, i.e., integrating the GPS-IR measurements of ground surface elevation changes, soil

moisture, and snow depth at one single GPS site in permafrost regions. We show a conceptual diagram of this framework in

Fig. 4. We first process the SNR interferograms during the data time span to obtain daily reflector heights, based on the steps

described in section 3.1. We then separate them into two groups, one in the snow season and the other in the snow-free

season. Reflector heights in the snow season are converted into snow depth. And the ones in snow-free days are used to

obtain ground surface elevation changes and the constant $H_0$. We then use the surface elevation changes and thermal indices

to model the surface deformation and calculate the time-varying $H_0'$. Lastly, we use the $H_0'$ and SNR data to estimate the

phases, which are converted to soil moisture content.




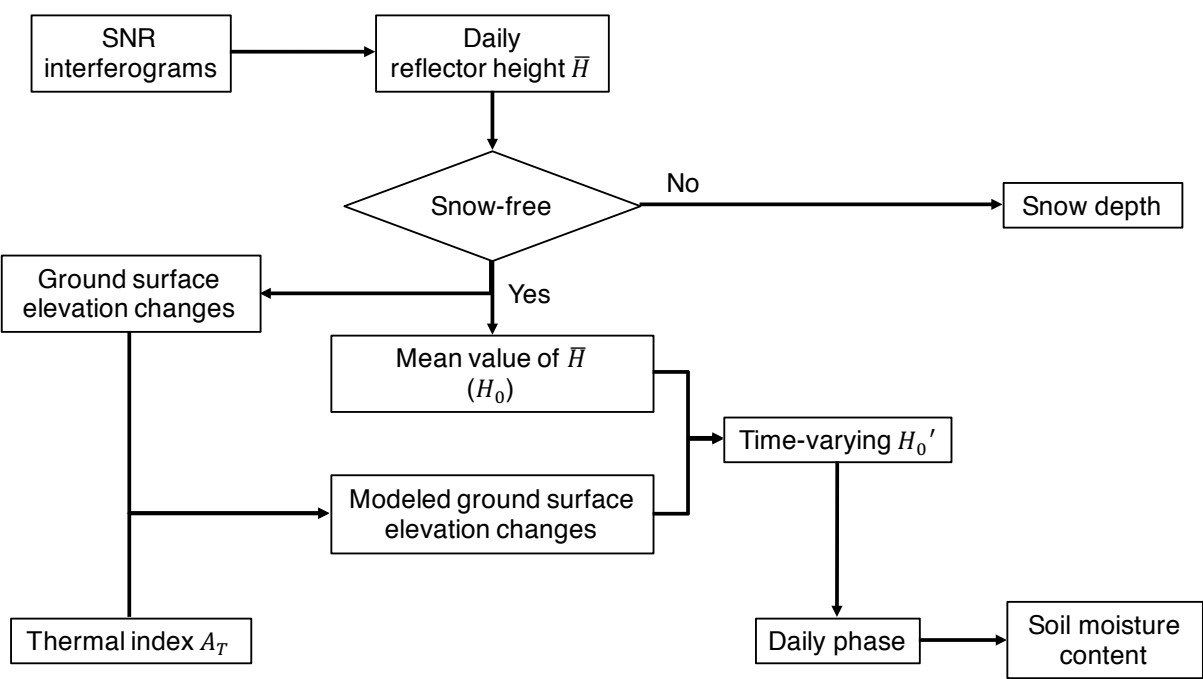

**Figure 4: Diagram of the 3-in-1 framework for integrating the GPS-IR measurements of ground surface elevation changes, soil moisture, and snow depth at one site.**

## 3.4 Data

SNR data of QLBG are available from November 2016 to April 2019 (Fig. 5). They have frequent large gaps, such as during

DOY 57–111 and DOY 191–215 in 2017 and DOY 77–180 in 2018, due to instrumentation problems. Considering the

continuity of SNR data and availability of in situ measurements, we use the SNR data of GPS L1 C/A signals during DOY

112–156 in 2017 (to obtain snow depth) and DOY 182–243 in 2018 (to measure ground surface elevation changes and soil

moisture content). The SNR data during 167–173 in 2017 (as the beginning of thaw season) are used to obtain the reflector

height of ground surface to be the reference for estimating snow depth. The sampling rate of SNR data is 15 s. SNR data are

recorded as integers. L2C signals are not recorded.

Snow depth was measured manually during DOY 112–156 in 2017. It was measured daily at ten points, randomly distributed

near the ultrasonic snow depth sensor in the east of QLBG (Fig. 1). We average them to obtain the daily measurements and

use the standard deviation of mean values as uncertainties. We use the manual observations to validate our GPS-IR

measurements. We do not use the ultrasonic measurements as they are not calibrated.

We also use in situ soil moisture and ground surface temperature in 2017 and 2018. We use the ground temperatures to

determine the onset and duration of thaw seasons. We also use them to calculate the thermal index $A_T$. We use the soil

moisture observations to convert phases to soil moisture content, and then to validate our modified GPS-IR method.

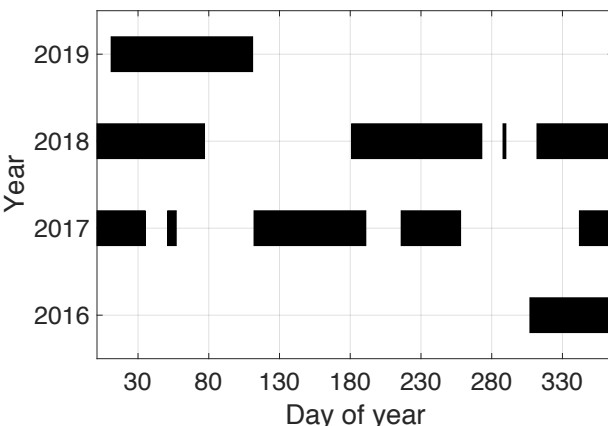

**Figure 5: Availability of the SNR data of QLBG during 2016–2019.**

## 4 Results

### 4.1 Ground surface elevation changes

We obtain the daily reflector height measurements during DOY 182–243 in 2018. And we remove the mean value from the

minus reflector heights and use the residuals to represent the ground surface elevation changes, which are shown in Fig. 6(a).

Their uncertainties are on the order of 1–2 centimeters. Surface elevation changes show a progressive subsidence trend,

which is mainly caused by ground ice melting within the thawing active layer.


We compare the surface elevation changes to the normalized thermal indices and build their best linear fit to obtain the

model parameters based on section 3.2.2 (Fig. 6(b)). The parameters $d_s$ and $d_0$ are -1.7 ± 0.8 cm and 1.2 ± 0.6 cm,

respectively. We then use these parameters and the normalized thermal indices to simulate the ground surface elevation

changes, which are presented in Fig. 6(a) as a curve superimposed on the GPS-IR measurements. The simulated surface

deformation is used to compute the time-varying $H_0'$, to estimate the phases and then soil moisture content, which will be

presented in section 4.2.

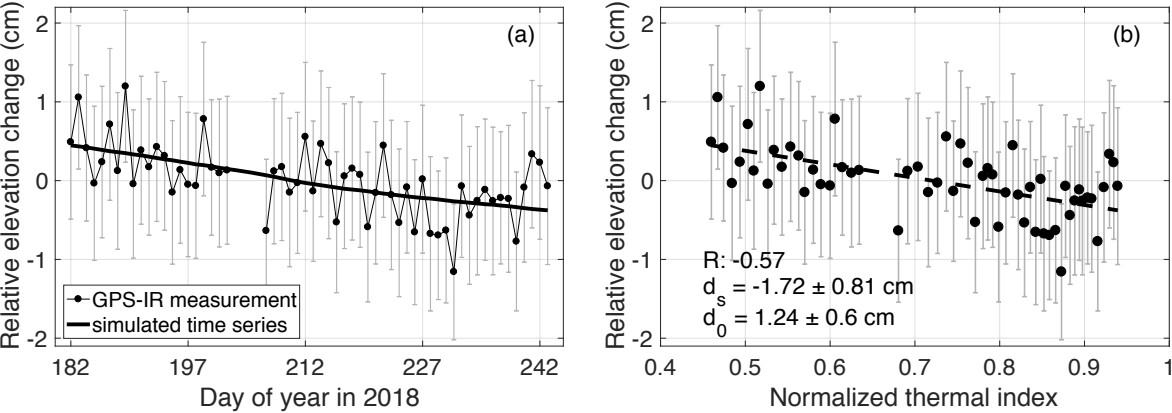

**Figure 6: (a) Time series of GPS-IR measurements of ground surface elevation changes and the simulated ones based on the model**

**parameters $d_s$ and $d_0$ in (b) during DOY 182–243 in 2018. (b) Scatter plot between the ground surface elevation changes and the**

**normalized thermal indices and their best linear fit (dashed line). $d_s$ and $d_0$ are the slope and intercept of this fit line.**

**4.2 Validation of the improved method and the soil moisture content**

We use the time-varying $H_0'$ as obtained in section 4.1 to estimate the daily phases during DOY 182–243 in 2018 (Fig. 7(a)).

For comparison, we also obtain the phases by the default method (Fig. 7(c)). We link the phases to in situ soil moisture

measurements to obtain the best linear fits as mapping functions. The fit line for the modified method has a slope of 1.73% ±

0.07% deg$^{-1}$ and an intercept of 22.2% ± 0.6% (Fig. 7(b)). The one for the default method has a slope of 0.84% ± 0.04% deg$^{-}$

$^1$ and intercept of 30% ± 0.39% (Fig. 7(d)). We use these parameters to convert the phases to soil moisture content shown in

Fig. 8.

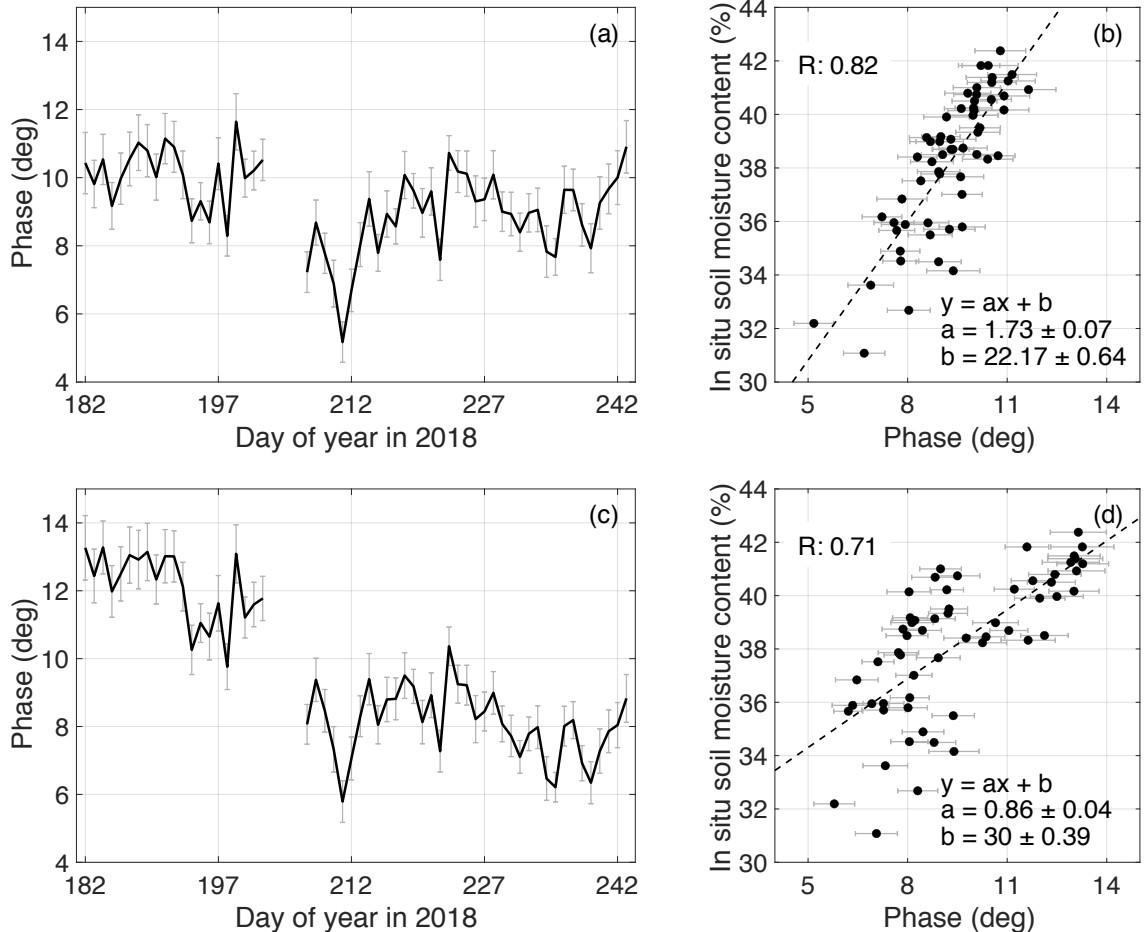

**Figure 7: (a) Time series of the phases derived by our method during DOY 182–243 in 2018. Error bars denote their uncertainties represented by the standard deviations of the mean values. (b) Scatter plot between phases in (a) and in situ soil moisture and their best linear fit shown as a dashed line. The slope and intercept of the fit line are presented. We also show the goodness of fit as R. (c) and (d) are similar to (a) and (b) correspondingly but for the results of the default method. The data gap during DOY 203–206 is due to the absence of SNR data.**

From the in situ measurements in Fig. 8, we observe that surface soil underwent several drying/wetting cycles during DOY 182–243 in 2018, especially the one during DOY 201–218 with a magnitude of ~10%. The soil moisture estimated by our method agrees well with the in situ ones (Fig. 8(a)). They capture the prominent drying/wetting feature and other cycles as well. The correlation coefficient between the in situ measurements and the estimates by our method is 0.82, and they have a





root mean squared error (RMSE) of 1.51%. In contrast, for the soil moisture derived by the default method, we barely

recognize the significant drying-wetting phenomenon during DOY 201–218. Furthermore, they exhibit an obvious

descending trend. Based on the simulations in section 3.2.1, the phase bias is nearly proportional to surface deformation and

surface subsidence introduces a negative bias. In summer, when the thawing front advances deeper, the surface subsidence

accumulates, then the bias decreases (with increasing absolute value). Accumulating surface subsidence gives rise to a

decreasing trend of soil moisture estimates. The soil moisture estimates by the default method and in situ measurements have

a correlation coefficient of 0.71 and RMSE of 1.85%. In summary, by comparing the GPS-IR estimates and in situ

observations, our method outperforms the default one.

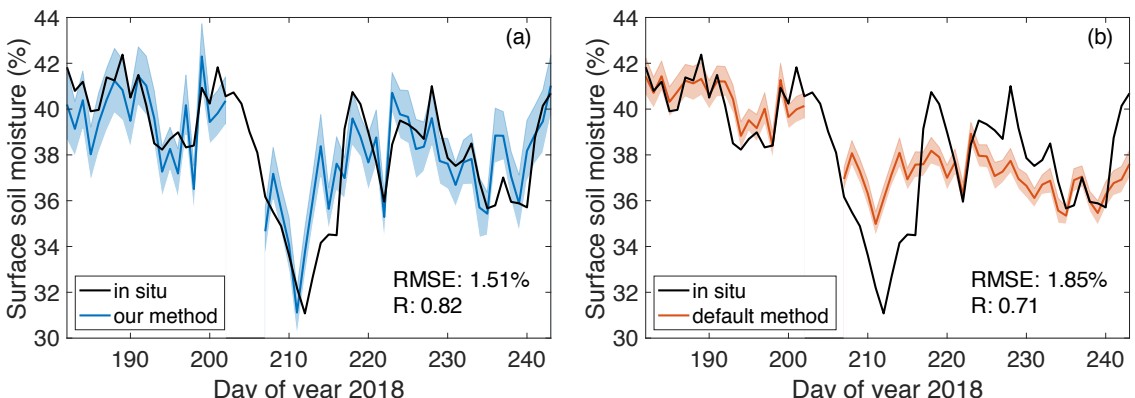


**Figure 8: Time series of in situ soil moisture content and those measured by (a) our method and (b) the default method. The shaded areas denote the uncertainties of GPS-IR estimates. Root-mean-square error (RMSE) and correlation coefficient (R) are presented.**

### 4.3 Snow depth

We obtain the GPS-IR-measured snow depth during DOY 112–156 in 2017 (Fig. 9). Their uncertainties are on the order of

several centimeters as well. We compare them to the manual observations. From the manual measurements, we can observe

that snow cover underwent several cycles of accumulation/ablation, e.g., the one during DOY 140–147. Consistently, GPS-

IR measurements capture these processes. Fig. 9(b) shows the scatter plot between the GPS-IR and manual observations and





their best linear fit. The correlation coefficient is 0.72. GPS-IR measurements have an RMSE of 7.57 cm and a bias of 6.52

cm.

The main reason for such a magnitude of bias and RMSE might be the inconsistency between the GPS-IR sensing area and

the probing positions. We use all SNR interferograms around QLBG, whose footprints are indicated in Fig 1(c). However,

the manual probing positions are randomly distributed to the east of QLBG. Moreover, snow and the ground have different

reflectivity for GPS signals. The reflectivity affects the amplitude and polarization of the reflected signals and then their

interference with the direct ones (Zavorotny et al., 2010). Thus, using the reflector height of the ground surface as a

reference may also introduce bias. Furthermore, possible penetration into the soil when manually probing the rod may also

introduce bias (McCreight et al., 2014). The QLBG's monument was expected to be stable during DOY 112–156 in 2017

when the ground was in a frozen state and would therefore have little impact on the GPS-IR estimates.


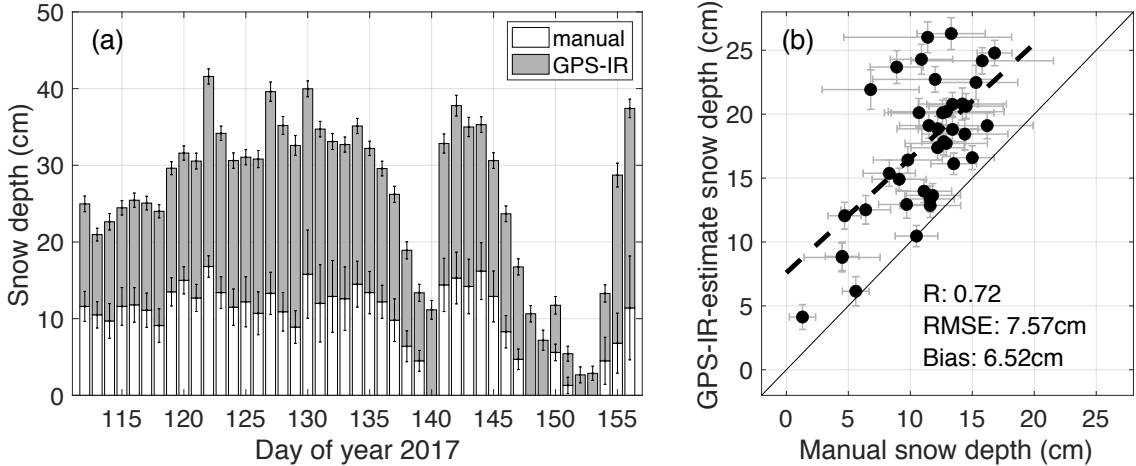

**Figure 9: (a) Stack bar plots of snow depth measured by GPS-IR and manually. Error bars denote their uncertainties. (b) Scatter plots of these two datasets and their best linear fit as a dashed line.**

# 5 Discussion

## 5.1 Merits and error sources of the modified method

The advantage of the modified method for estimating soil moisture content is that it can correct the bias introduced by the seasonal surface elevation changes in permafrost areas. Moreover, the required input of this method is easy to access. The surface elevation changes can be obtained directly by GPS-IR. The ground/air temperatures can be provided by in situ measurements or reanalysis datasets (e.g., European Centre for Medium-Range Weather Forecasts, https://www.ecmwf.int).

In addition, this modified method can also provide daily and continuous measurements with intermediate spatial coverages.

The error sources of the modified method are related to the antenna gain pattern, surface conditions (i.e., vegetation and soil moisture), and the GPS-IR geometry. The receiver antenna is designed to favor the reception of direct signals and suppress those with low satellite elevation angles and reflected signals. It has asymmetric gain pattern along the elevation angle. The

surface reflectivity also varies with the elevation angle. Therefore, the phases of each point of a given SNR interferogram are slightly different (i.e., the phase term $\phi(e)$ in equation (1) is a variable). Assuming the phase as a constant in data processing might introduce errors. Zavorotny et al. (2010) simulated SNR in bare soil conditions with a smooth reflecting surface. The phase variations with respect to the elevation angles are nearly the same, given a change of soil moisture content. It indicates that assuming the phase as a constant has little impact on the GPS-IR estimates of soil moisture, as we focus on the temporal

changes of the phase. It also implies that antenna gain pattern has little impact on the GPS-IR measurements. The ground surface at our site is relatively smooth and horizontal (Fig. 1). The vegetation is sufficiently short (i.e., less than the wavelength of the L-band GPS signals) to be transparent for the GPS signals. The impact of the surface conditions is expected to be limited.

Regarding the error sources related to the GPS-IR geometry, they affect the soil moisture estimates through the GPS-IR-measured surface elevation changes, which are used to calculate the time-varying $H_0'$. The error sources are mainly monument stability, tropospheric delays of the GPS signals, monument thermal contraction/expansion. Though the foundation base is slightly shallower than the active layer thickness, the monument does not have any significant



displacement pattern (see the details in the supplementary). The monument can be regarded as stable with respect to the

permafrost and barely has any impact on the GPS-IR-measured surface elevation changes. Regarding the tropospheric

delays, we use in situ air temperature and pressure measurements to quantify them, using the refraction correction model of

Bennett (1982). The tropospheric biases are ~1.3 cm and relatively steady (Fig. 10). As we focus on the temporal changes of

the surface elevation changes, the impact of the tropospheric biases is negligible. For the thermal expansion/contraction of

the monument, the coefficient of linear thermal expansion of galvanized steel is $11\sim13 \times 10^{-6}$ m·(m·°C)$^{-1}$. Given a

temperature variation range of 20 °C in a thaw season, for a 2-m-high monument, the magnitude of the thermal expansion is

less than 1 mm, at least one order of magnitude smaller than that of surface elevation changes.

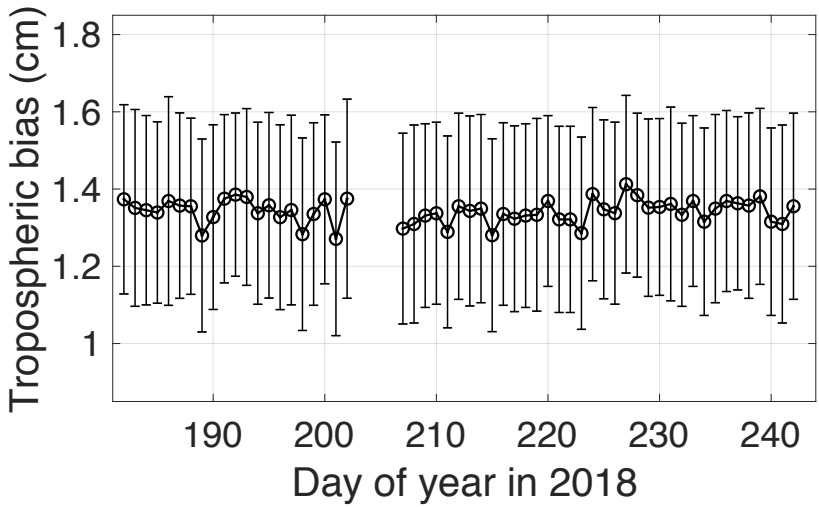

**Figure 10: Tropospheric biases of ground surface elevation changes during DOY 182–243 in 2018. They are the mean values of the**

**biases of each satellite tracks, whose standard deviations are represented by error bars.**

The temporal variation of soil moisture may also introduce bias to the GPS-IR-measured surface elevation changes then to

the measurements of soil moisture content. As the phase variations with respect to the elevation angles are not exactly the

same given a change of soil moisture, assuming the phase as a constant introduces bias to the frequency retrievals and then

reflector heights. Such bias is called compositional reflector height, as it manifests itself as a part of reflector height

(Nievinski, 2013). Liu and Larson (2018) conducted simulations and found that the bias was less than 2 cm and varied in the

range of less than 1 cm, given soil moisture between 15% and 40%. In this study, such bias is expected to be limited, as the precipitation is scarce and light in the cold and dry plateau climate and that we focus on the temporal changes of the surface elevations.

**5.2 Benefits of the three-in-one framework to permafrost studies**

The three-in-one framework can fully utilize the GPS-IR in permafrost studies. We can obtain the GPS-IR measurements of three key variables by one GPS station, which is crucially important due to the lack of observations in permafrost areas. Our study in QLBG serves as an example of this framework.

The obtained GPS-IR measurements can be directly used to study the changes in permafrost areas at local scales. We can use the estimates of snow depth to study the variation of snow cover and its insulating effect and hydrological impact on the frozen ground. The ground surface elevation changes can indicate the amount of melting ground ice and the changes of active layer thickness with ancillary information such as soil moisture profiles. The magnitude of surface elevation changes in permafrost areas mainly depends on the moisture content within the thawed/frozen soil. We can estimate the active layer 390 thickness if we have the seasonal surface elevation changes and soil moisture profile (Liu et al., 2012). The soil moisture content measurements aid in studying the water cycles and surface energy balance and analyzing the interaction between the atmosphere and the frozen ground.

The GPS-IR measurements can fill some spatial gaps. For instance, in the QTP, the permafrost is extensive but with limited 395 and unevenly distributed monitoring sites. Most of them are located along the QTP highway and railway (Zou et al., 2017). QLBG in our study can fill a spatial gap, providing GPS-IR measurements to detecting the changes of frozen ground. Moreover, the existing monitoring sites usually provide records of ground temperature and active layer thickness. The GPS-IR measurements can be complementary to them.

The GPS-IR measurements can be used to calibrate/validate the remote sensing observations at regional scales. At present, surface elevation changes, soil moisture, snow depth can be provided by air/satellite-borne measurements, such as InSAR
measurements for surface elevation changes (Liu et al., 2010), Soil Moisture Active Passive (SMAP) data for soil moisture content (Entekhabi et al., 2010), and snow depth estimated from passive microwave radiometry data (Walker and Silis, 2002; Che et al., 2008),  These remote sensing observations have relatively broad spatial coverages, typically on the order of

several tens of kilometers. As the GPS-IR measurements have intermediate spatial coverages, they can be used to calibrate/validate the remote sensing observations.

The three-in-one framework can be extended to other GPS sites, such as those in the Arctic. The GPS station SG27 in Utqiagvik (formerly Barrow), Alaska has been proved to be usable for GPS-IR studies (Liu and Larson, 2018). Zhang et al.

(2020) also identified 12 usable GPS stations in the Northern Canada permafrost areas. We can apply the framework to these stations to obtain GPS-IR measurements, which will contribute to the research on permafrost.

## 6 Conclusion

This study highlights the improvement to the default GPS-IR algorithm for estimating soil moisture content. It can correct the bias introduced by the seasonal surface elevation changes in permafrost areas. We use the GPS data and the in situ

measurements of soil moisture at QLBG to validate this modified method. The correlation coefficient and RMSE between the GPS-IR estimates and the in situ ones improve from 1.85% to 1.51% and 0.71 to 0.82, respectively.

We implement a framework to integrate the GPS-IR measurements of ground surface elevation changes, soil moisture, and snow depth at one single site. Following the framework, we obtain the GPS-IR measurements at QLBG. We also validate the

GPS-IR-measured snow depth by manual observations. They have a correlation coefficient of 0.72, an RMSE of 7.57 cm, and a bias of 6.52 cm. The framework helps to comprehensively use GPS-IR in frozen ground. It also can be extended to other sites, for instance, those in the Arctic, where multiple stations have been identified to be usable for GPS-IR studies.

This study is also the first to use GPS-IR in the QTP. QLBG fills a spatial gap in the existing sparse permafrost-monitoring

sites. Its GPS-IR measurements are complementary to the existing observations, such as ground temperatures. They can also

be used to calibrate/validate remote sensing observations.

**Code and data availability**

We will publish our data in a publicly accessible database such as PANGAEA.

**Author contribution**

JZ conducted the data processing and result analysis and wrote the manuscript. LL helped to interpret the results and revised the manuscript. TC and LS provided the GPS data and in situ measurements and revised the manuscript.

**Competing interest**

The authors declare that they have no conflict of interest.

**Acknowledgments**

We thank Dr. Yufeng Hu for the constructive discussion.

**Financial support**

This research was supported by the Second Tibetan Plateau Scientific Expedition and Research Program (STEP) (Grant No. 2019QZKK0201), the Strategic Priority Research Program of the Chinese Academy of Sciences (Grant No. XDA19070204), and the Hong Kong Research Grants Council (Grant no. CUHK14305618).

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
