# Peer review of "Three-in-one: GPS-IR measurements of ground surface elevation changes, soil moisture, and snow depth at a permafrost site in the northeastern Qinghai-Tibet Plateau"

_The Cryosphere, 2020_

## Referee Comment (RC1) · Anonymous Referee #1 · 16 Sep 2020

General comments: This paper presents a modified GPS-IR algorithm for soil moisture retrieval in permafrost regions. The authors find that the modified algorithm performs slightly better than the default method, which can be seasonally-biased due to surface deformation changes resulting from the thawing active layer. The authors find a significant bias between their GPS-IR snow depth retrievals and in situ observations near the antenna.

Overall, I found the paper to be well written with clear objectives. Since the results presented in this paper are only from one GPS antenna with a limited time series, they

are preliminary, and it is difficult to know if the methodology will be successful at other GPS monuments. However, it makes for a good initial study in this topic.

Specific comments: Figure 2: I think the antenna anchoring position should be drawn not in the permafrost, but in the active layer, since that is where your antenna monument is anchored.

Line 160: Why do you only use 5-15 degrees in this study, when other GPS-IR studies more commonly use 5-30 degrees?

Line 167: Reflector height does have a dependence on soil moisture, though it is not as linear as the dependence of phase on soil moisture. How do you know that your summertime reflector height variations are due to ground surface elevation changes and not due to soil moisture?

Line 323: You say that one reason for the difference between in situ snow depth observations and snow depths derived from GPS-IR is the difference in sampling areas. Couldn't you check this by excluding the SNR observations that lie outside the manual probing positions?

---

## Referee Comment (RC2) · Anonymous Referee #2 · 29 Sep 2020

This is a review three-in-one: GPS-IR measurements of ground surface elevation changes, soil moisture, and snow depth at a permafrost site in the northeastern Qinghai-Tibet Plateau by Jiahua Zhang et al. The authors used one GNSS station to estimate soil moisture, snow depth, and ground surface elevation changes. They used the estimated ground surface elevation to improve the soil moisture estimates. Although GNSS-IR looks very promising these days, the manuscript in the current format is rejected and it is not recommended for the publication. Larson et al. (2019) used several stations from the PBO network and reported soil moisture, snow depth,

vegetation water content, and water loading. It should be noted that the reported soil moisture estimates from 11 GNSS sites by Small et al. (2016) look way better than what is reported in this manuscript for the soil moisture. In addition, ground surface elevation changes over the permafrost area using GNSS-IR were already reported by Liu at al. (2018). The reported bias for the snow depth estimates is also much larger than what is already reported, i.e. Larson et al. (2016); Siegfried et al. (2017). Therefore, the authors should explicitly answer which challenge/challenges of GNSS-IR is tackled where recent publications have demonstrated even way better results. In addition, it is very hard to assess the results qualitatively using just one single GNSS station. The authors also discard the impact of the GNSS antenna by using the gin pattern of TRM29659.00.

---

## Referee Comment (RC3) · Anonymous Referee #3 · 23 Feb 2021

This paper introduces GPS interferometric reflectometry technique as a tool for remote sensing of surface elevation changes, soil moisture contents, and snow depth at a single permafrost site. In addition, authors proposed an improved method for soil moisture estimation by modeling surface vertical movements and removing its bias on reflected GPS SNR phase shifts.

Overall, the objectives and approaches are clear and the proposed solution for soil moisture estimation at permafrost areas with vertical displacements is a genius idea. However, there are some concerns which convince me to call for a major revision for

this paper.

1. The inconsistency between GPS-IR-driven and in-situ-measured snow depth values is out of acceptable range. GPS-IR capability of snow depth measurement has been examined several times in many studies, and strong agreements have been achieved. Although the correlation in this paper looks promising and reflects the general patterns of snow accumulation, the bias is not acceptable as previous studies have reached to better agreements. In addition, the way that the authors explained this "systematic" inconsistency does not make sense. The reflectivity difference between snow and the underlying frozen ground is not so much for GPS L-band signals. Moreover, this reflectivity difference, even if we consider it as a potential source of error, would not affect neither the amplitude nor the polarization because signals are assumed to be reflected off the "top" of the snowpack. Furthermore, "possible penetration into the soil when manually probing the rod", if happened, would reduce the bias as it would cause an overestimation in in-situ snow depth measurements. I would seek for either a better explanation or a reconsideration in the snow depth retrieval method. Looking into "higher-order frequencies" can be a solution for this issue as proposed by Cardellach, Fabra, et al. (2012) and Ghiasi (2020).

2. The authors have used Stefan's equation for modeling the surface elevation changes as they believe GPS-IR elevation retrievals are not accurate enough because their uncertainties are in the order of few centimetres. I would say that "a few centimetres" is an acceptable accuracy for this purpose since Stefan's equation has not shown a better accuracy in literature. I would suggest conduct the same validation using surface elevations directly obtained by GPS-IR. Besides, the term "uncertainty" used by the authors does not look very exact because it is driven based on the standard deviation of the mean values which are not necessarily to be normally distributed.

3. Although the paper appears in a very clear and accurate English writing, some sentences are too short, e.g., line 221, and some sentences start with "And" which looks somehow inappropriate in academic English writing, e.g., lines 236 and 266.

---

## Author Comment (AC1) · 13 Apr 2021

**Responses to RC1**

General comments: This paper presents a modified GPS-IR algorithm for soil moisture retrieval in permafrost regions. The authors find that the modified algorithm performs slightly better than the default method, which can be seasonally-biased due to surface deformation changes resulting from the thawing active layer. The authors find a significant bias between their GPS-IR snow depth retrievals and in situ observations near the antenna.

Overall, I found the paper to be well written with clear objectives. Since the results presented in this paper are only from one GPS antenna with a limited time series, they are preliminary, and it is difficult to know if the methodology will be successful at other GPS monuments. However, it makes for a good initial study in this topic.

We thank the reviewer for his/her constructive comments. We have addressed all of them with point-by-point replies given below. The line numbers refer to the previously submitted discussion paper, aiming to point out where the discussion paper has partly addressed the reviewer's comments.

Specific comments:

1. Figure 2: I think the antenna anchoring position should be drawn not in the permafrost, but in the active layer, since that is where your antenna monument is anchored.

We have revised Figure 2 to draw the anchoring position near the active layer base (Fig. R1).

[Figure]

Figure R1: Diagrams showing the geometries of GPS-IR in the thawing/freezing active layer conditions. The symbols $V$, $S$, and $P$ denote the vertical positions of the GPS antenna, reflecting surface, and the monument anchor point with respect to an arbitrary reference in the deep permafrost, respectively. $H$ denotes the reflector height. $d$ represents the surface deformation due to active layer freezing/thawing.

2. Line 160: Why do you only use 5-15 degrees in this study, when other GPS-IR studies more commonly use 5-30 degrees?

In the framework of GPS-IR, one of the bases is that SNR observations should have a clear sinusoidal pattern. Then the SNR metrics of frequency, phase, and amplitude can be utilized to estimate environmental variables. In Fig. R2, we present a piece of SNR observations at the elevation angles of 5–30 degrees recorded by our GPS station as an example. The SNR observations at 5–15 degrees show a clear sinusoidal pattern. However, they are irregular from 15 to 30 degrees, thus we choose not to use them in our analysis. Since the sampling rate of SNR data is 15 s, we still have enough data points at 5–15 degrees to determine the dominant frequency and phase.

[Figure]

Figure R2: SNR observations with elevation angles of 5–30 degrees of the GPS-02 satellite on DOY 220 in 2018. The second-order polynomial has been removed.

3. Line 167: Reflector height does have a dependence on soil moisture, though it is not as linear as the dependence of phase on soil moisture. How do you know that your summertime reflector height variations are due to ground surface elevation changes and not due to soil moisture?

Yes, we agree that reflector height can be affected by soil moisture. For a given SNR series, phases of the data points are not the same, but slightly vary with respect to elevation angle (i.e., $\phi(e)$ is not a constant but varies with elevation angle). In data processing, treating the phase as a constant introduces an error into the reflector height, which is usually called as compositional reflector height (Nievinski, 2013).

The compositional reflector height can be simulated by a multipath model from Nievinski and Larson (2014). As the gain pattern of the GPS antenna at our site is not available, we alternatively use the one of TRM29659.00 with a radome of SCIT to run simulations. We also use the daily in situ soil moisture observations at our site as the input. The simulated results are presented in Fig. R3. The parameters for the simulations are shown in Table R1.

In Fig. R3, the simulated compositional heights are within the range of 2–2.7 cm. The peak-to-peak magnitude of their variations is 0.7 cm. This is less than the magnitude of the uncertainty of reflector height retrieval, which is on the order of several centimeters. As we focus on the

temporal variations of the retrieved reflector heights (not the absolute heights), the soil moisture's impact is limited.

Furthermore, we observe a decreasing trend from the compositional heights with a rate of 0.0039 cm/day, which leads to a surface uplift (as surface elevation changes are the reverses of reflector height variations) and makes the GPS-IR measurements underestimate surface subsidence. However, the introduced uplift has a negligible magnitude. The overall uplift is only 0.24 cm during DOY 182–243 in 2018, which barely affects the clear subsidence trend presented by the GPS-IR measurements (*Page 13, Lines 265–281*).

Given the limited magnitude of compositional height variations and its negligible decreasing trend (causing surface uplift), the reflector height variations are mainly caused by surface elevation changes due to thawing active layer rather than soil moisture changes.

[Figure]

Figure R3: (a) In situ surface soil moisture during DOY 182–243 in 2018 at our study site. (b) Simulated compositional reflector height by using the in situ soil moisture. The y axis has been reversed to make the variations of compositional height consistent with surface vertical movement. The dashed line is the best linear fit of the simulations, whose slope is presented as well.

Table R1: Key parameters used in the simulations

| Parameter | Value |
| --- | --- |
| Antenna (Radome) | TRM29659.00 (SCIT) |
| Signal | L1 C/A |
| Antenna height | 2 m |
| Azimuth angle range | 0–360° |
| Elevation angle range | 5–15° |

| Reflector material | Sandy loam |
|---|---|
| Soil moisture content | Daily in situ measurements during DOY 182–243 in 2018 |

4. Line 323: You say that one reason for the difference between in situ snow depth observations and snow depths derived from GPS-IR is the difference in sampling areas. Couldn't you check this by excluding the SNR observations that lie outside the manual probing positions?

The manual probing positions are located in the southeast direction of the GPS station and close to the automatic snow sensor, which are generally within the azimuth range of 90–135 degrees. We now use the SNR data within this azimuth range to calculate snow depth, which are presented in Fig. R4. We can observe that the new GPS-IR measurements have a better agreement with the in situ observations. The correlation coefficient, RMSE, and bias are 0.73, 4.11 cm, and 2.49 cm, respectively, for the new GPS-IR estimates (Fig. R4(c) and (d)), whereas 0.72, 7.57 cm, and 6.52 cm for the reported ones in the discussion paper (Fig. R4(a) and (b)). In the review paper of Larson (2019), the agreement level of the GPS-IR measurements to the in situ ones in their validation experiments ranges from 4 cm to 6 cm. Our new GPS-IR measurements show a comparable accuracy.

We have updated the results of snow depth by using the azimuth range of 90°–135° in our revised manuscript.

[Figure]

Figure R4: (a) Bar plots of snow depth measured manually or by GPS-IR during DOY 112–156 in 2017. The GPS-IR results are derived by using the SNR observations within the azimuth range of 0–360°. (b) Scatter plot between the manually measured and GPS-IR-retrieved snow depth. The correlation coefficient (R), root-mean-square error (RMSE), and bias are shown as well. (c) and (d) are similar plots to (a) and (b) correspondingly but for the GPS-IR results by using the SNR observations within 90°–135° azimuth range.

References:

Larson, K. M.: Unanticipated Uses of the Global Positioning System, Annu. Rev. Earth Planet. Sci., 47(1), 19–40, doi:10.1146/annurev-earth-053018-060203, 2019.

Nievinski, F. G.: Forward and Inverse Modeling of GPS Multipath for Snow Monitoring., 2013.

Nievinski, F. G. and Larson, K. M.: An open source GPS multipath simulator in Matlab/Octave, GPS Solut., 18(3), 473–481, doi:10.1007/s10291-014-0370-z, 2014.

---

## Author Comment (AC2) · 13 Apr 2021

Responses to RC2

This is a review three-in-one: GPS-IR measurements of ground surface elevation changes, soil moisture, and snow depth at a permafrost site in the northeastern Qinghai-Tibet Plateau by Jiahua Zhang et al. The authors used one GNSS station to estimate soil moisture, snow depth, and ground surface elevation changes. They used the estimated ground surface elevation to improve the soil moisture estimates. Although GNSS-IR looks very promising these days, the manuscript in the current format is rejected and it is not recommended for the publication.

We thank the reviewer for his/her constructive comments. We have addressed all of them with our point-by-point replies given below. The line numbers refer to the previously submitted discussion paper, aiming to point out where the discussion paper has partly addressed the reviewer's comments.

1. Larson et al. (2019) used several stations from the PBO network and reported soil moisture, snow depth, vegetation water content, and water loading.

The key difference between our study and Larson (2019) is that our study is dedicated for permafrost studies. We obtain surface elevation changes, surface soil moisture, and snow depth at the same GPS site. These three variables are all closely linked with frozen ground dynamics and directly pertinent to the readership of TC (*Page 2, Lines 40–57*).

The main contributions of this study have been explicitly stated in the discussion paper as: (1) the improvement of the default GPS-IR algorithm for estimating soil moisture in permafrost areas to correct the impact of seasonal surface elevation changes; (2) the GPS-IR measurements at a site in the Qinghai-Tibet Plateau (QTP); and (3) the three-in-one framework to fully utilize the potential of GPS-IR in permafrost studies (*Page 1, Lines 24–28*).

The significance of this study: The commonly used GPS-IR algorithm for soil moisture estimation does not consider the typical seasonal surface vertical movement in permafrost areas. Our improved method can remove the errors caused by seasonal surface deformation and estimate

soil moisture reliably in permafrost areas. Permafrost coverage is extensive in the QTP but with scarcely and unevenly distributed monitoring sites. Our study site, one of the first of this kind in the QTP, can fill a critical spatial gap and may raise the community's interest to set up more sites likewise. Moreover, our GPS-IR measurements can be supplementary to the permafrost temperature and provide new insights into frozen ground dynamics. The GPS-IR measurements can also calibrate/validate remote sensing observations (Zhang et al., 2020). The three-in-one framework integrates the measurements of these three variables at the same site, which can fully utilize the potentials of GPS-IR in permafrost studies (*Page 4, Line 81–92*).

2. It should be noted that the reported soil moisture estimates from 11 GNSS sites by Small et al. (2016) look way better than what is reported in this manuscript for the soil moisture.

Small et al. (2016) estimates soil moisture content at 11 sites with various vegetation conditions. They use different strategies at different sites based on vegetation water content. For individual sites, by using the optimal algorithms, their RMSEs range from 1.5% to 5.1% (Table R1). In our study, the RMSE by using our improved method is 1.5%. It indicates that the accuracy of our results is better than those at most of the sites in Small et al. (2016).

In our study, we do not consider the vegetation impact because the vegetation at our site (and many GNSS sites in continuous permafrost areas) is very short and relatively sparse, which is nearly transparent for the L-band GPS signals.

Table R1: The root-mean-square error (RMSE) of GPS-IR-estimated soil moisture content at various sites with different vegetation water content (Source: Small et al., 2016).

| Site | RMSE | Vegetation Water Content (kg·m$^{-2}$) |
| --- | --- | --- |
| P036 | 1.5% | 0.65 |
| P037 | 2.7% | No data |
| P038 | Not available | No data |
| P039 | Not available | 0.33 |

| P040 | Not available | 0.47 |
|------|--------------|------|
| P070 | 3% | 0.41 |
| P123 | Not available | 0.24 |
| MFLE | 3.5% | 0.32 |
| OKL2 | 3.8% | 1.02 |
| OKL3 | 3.6% | 1.01 |
| OKL4 | 5.1% | 1.40 |

3. In addition, ground surface elevation changes over the permafrost area using GNSS-IR were already reported by Liu and Larson (2018).

This study is the first one using GPS-IR to measure ground surface elevation changes at a site in the QTP. Permafrost is extensive in the QTP but the monitoring sites, such as boreholes, are scarce and unevenly distributed. Most of the monitoring sites are along the Qinghai-Tibet highway and railway. The GPS site in our study can fill a spatial gap in the QTP. The GPS-IR measurements can be supplementary to the existing ground temperature observations to provide insights into frozen ground dynamics in a new perspective.

The GPS-IR-estimated surface elevation changes are also the basis of our solution to improve the default algorithm for estimating soil moisture in permafrost areas. Furthermore, they are also the indispensable component of the three-in-one framework to fully utilize GPS-IR in permafrost studies.

Liu and Larson (2018) did not estimate snow depth or soil moisture.

4. The reported bias for the snow depth estimates is also much larger than what is already reported, i.e. Larson et al. (2016); Siegfried et al. (2017). Therefore, the authors should explicitly answer which challenge/challenges of GNSS-IR is tackled where recent publications have demonstrated even way better results.

The dominant reason for the relatively large bias is the inconsistency of sampling areas of GPS-IR and manual probing locations. The GPS-IR measurements are derived by using the SNR data within the azimuth range of 0–360°. Whereas, the manual probing locations are generally in the area with azimuth angle of 90°–135°.

We use the SNR data within the azimuth range of 90°–135° to calculate snow depth, which are presented in Fig. R1. We can observe that the new GPS-IR measurements have a better agreement with the in situ observations. The correlation coefficient, RMSE, and bias are 0.73, 4.11 cm, and 2.49 cm, respectively, for the new GPS-IR estimates (Fig. R1(c) and (d)), whereas 0.72, 7.57 cm, and 6.52 cm, respectively, for the reported ones in the discussion paper (Fig. R1(a) and (b)). Our new GPS-IR results have comparable quality with those of Larson and Small (2016) and Siegfried et al. (2017).

We have updated the results of snow depth by using the azimuth range of 90°–135° in our revised manuscript.

[Figure]

Figure R1: (a) Bar plots of snow depth measured manually and by GPS-IR using SNR within the 0–360° azimuth range. (b) Scatter plot of the manual measured and GPS-IR-estimated snow depth. The correlation coefficient (R), root-mean-square error (RMSE), and bias are presented. (c) and (d) are similar to (a) and (b), correspondingly, but for the GPS-IR measurements over the azimuth range of 90°–135°.

We have reiterated and demonstrated our contributions in reply to comment #1. The highlight of this paper is the improvement of the default algorithm for estimating soil moisture in permafrost areas. Our method can correct the impact of seasonal surface elevation changes and improve the accuracy of results. Another contribution is the three-in-one framework, integrating the GPS-IR measurements of snow depth, surface elevation changes, and soil moisture at one site. Our snow depth result is the necessary component of the framework implementation. The three-in-one framework can fully utilize the potential of GPS-IR in studying permafrost. Moreover, this study is also the first one reporting GPS-IR measurements in the QTP.

4. In addition, it is very hard to assess the results qualitatively using just one single GNSS station.

To validate the improved method, we need a suitable GPS site and in situ soil moisture content within 0–5 cm layer in permafrost areas. However, it is challenging to find such a site because most of the existing GPS sites are designed and maintained for monitoring solid earth movement and ionospheric variations. Soil moisture observations are usually absent at these sites. At our site, it is fortunate to have a co-located integrated weather station recording soil moisture content for assessing and validating our method. Though only using one site, the proof of concept of improving the default algorithm for estimating soil moisture content in permafrost areas has been clearly presented. Though only using one site, we have not only presented the concept the improved method but proved its effectiveness by comparing the GPS-IR estimates and in situ observations.

5. The authors also discard the impact of the GNSS antenna by using the gain pattern of TRM29659.00.

The gain pattern of TRM29659.00 is used to simulate SNR observations by using the model of Nievinski and Larson (2014). At our study site, the antenna gain pattern of QLBG is not available. We alternatively use the one of TRM29659.00.

The impacts of gain pattern on the simulated phase bias and the GPS-IR results are negligible. The gain pattern is usually designed to be nonuniform along elevation angle, to favor the reception of the direct signals and suppress the reflected. For any given SNR series, the antenna gain's impact varies at each data point. But for any pair of SNR series with the same elevation angles, they suffer the same impact from the gain pattern. Therefore, the influence of antenna gain pattern on the SNR metrics (i.e., frequency, amplitude, phase) can be regarded as a systematic bias. To sum up, as we focus on the temporal variations, the impact of antenna gain pattern is negligible.

References:

Larson, K. M. and Small, E. E.: Estimation of Snow Depth Using L1 GPS Signal-to-Noise Ratio Data, IEEE J. Sel. Top. Appl. Earth Obs. Remote Sens., 9(10), 4802–4808, doi:10.1109/JSTARS.2015.2508673, 2016.

Larson, K. M.: Unanticipated Uses of the Global Positioning System, Annu. Rev. Earth Planet. Sci., 47(1), 19–40, doi:10.1146/annurev-earth-053018-060203, 2019.

Liu, L. and Larson, K. M.: Decadal changes of surface elevation over permafrost area estimated using reflected GPS signals, Cryosph., 12(2), 477–489, doi:10.5194/tc-12-477-2018, 2018.

Nievinski, F. G. and Larson, K. M.: An open source GPS multipath simulator in Matlab/Octave, GPS Solut., 18(3), 473–481, doi:10.1007/s10291-014-0370-z, 2014.

Siegfried, M. R., Medley, B., Larson, K. M., Fricker, H. A. and Tulaczyk, S.: Snow accumulation variability on a West Antarctic ice stream observed with GPS reflectometry, 2007-2017, Geophys. Res. Lett., 44(15), 7808–7816, doi:10.1002/2017GL074039, 2017.

Small, E. E., Larson, K. M., Chew, C. C., Dong, J. and Ochsner, T. E.: Validation of GPS-IR Soil Moisture Retrievals: Comparison of Different Algorithms to Remove Vegetation Effects, IEEE J. Sel. Top. Appl. Earth Obs. Remote Sens., 9(10), 4759–4770, doi:10.1109/JSTARS.2015.2504527, 2016.

Zhang, J., Liu, L. and Hu, Y.: Global Positioning System interferometric reflectometry (GPS-IR) measurements of ground surface elevation changes in permafrost areas in northern Canada, Cryosph., 14(6), 1875–1888, doi:10.5194/tc-14-1875-2020, 2020.

---

## Author Comment (AC3) · 13 Apr 2021

Responses to RC3

This paper introduces GPS interferometric reflectometry technique as a tool for remote sensing of surface elevation changes, soil moisture contents, and snow depth at a single permafrost site. In addition, authors proposed an improved method for soil moisture estimation by modeling surface vertical movements and removing its bias on reflected GPS SNR phase shifts.

Overall, the objectives and approaches are clear and the proposed solution for soil moisture estimation at permafrost areas with vertical displacements is a genius idea. However, there are some concerns which convince me to call for a major revision for this paper.

We thank the reviewer for his/her constructive comments. We have addressed all of them with our point-by-point replies given below. The line numbers refer to the previously submitted discussion paper, aiming to point out where the discussion paper has partly addressed the reviewer's comments.

1. The inconsistency between GPS-IR-driven and in-situ-measured snow depth values is out of acceptable range. GPS-IR capability of snow depth measurement has been examined several times in many studies, and strong agreements have been achieved. Although the correlation in this paper looks promising and reflects the general patterns of snow accumulation, the bias is not acceptable as previous studies have reached to better agreements. In addition, the way that the authors explained this "systematic" inconsistency does not make sense. The reflectivity difference between snow and the underlying frozen ground is not so much for GPS L-band signals. Moreover, this reflectivity difference, even if we consider it as a potential source of error, would not affect neither the amplitude nor the polarization because signals are assumed to be reflected off the "top" of the snowpack. Furthermore, "possible penetration into the soil when manually probing the rod", if happened, would reduce the bias as it would cause an overestimation in in-situ snow depth measurements. I would seek for either a better explanation or a reconsideration in the snow depth retrieval method. Looking into "higherorder frequencies" can be a solution for this issue as proposed by Cardellach, Fabra, et al. (2012) and Ghiasi (2020).

We divide this comment into four points and address them accordingly.

1.1 The inconsistency between GPS-IR-driven and in-situ-measured snow depth values is out of acceptable range. GPS-IR capability of snow depth measurement has been examined several times in many studies, and strong agreements have been achieved. Although the correlation in this paper looks promising and reflects the general patterns of snow accumulation, the bias is not acceptable as previous studies have reached to better agreements.

The dominant reason for the relatively large bias of our reported GPS-IR snow depth is the inconsistency of the sampling areas. The spatial coverage of the GPS-IR observations is around the GPS station. Whereas, the probing positions are generally in the southeast direction of the station and close to the automatic snow sensor, which is generally within the azimuth range of 90°–135°.

We now use the SNR data within the azimuth range of 90°–135° to calculate snow depth, which are presented in Fig. R1. We can observe that the new GPS-IR measurements have a better agreement with the in situ ones. The correlation coefficient, RMSE, and bias are 0.73, 4.11 cm, and 2.49 cm, respectively, for the new GPS-IR observations (Fig. R1(c) and (d)), whereas 0.72, 7.57 cm, and 6.52 cm, respectively, for the reported ones in the discussion paper (Fig. R1(a) and (b)). In the review paper of Larson (2019), the agreement level of the GPS-IR measurements to the in situ ones in their validation experiments ranges from 4 cm to 6 cm. Our new GPS-IR measurements show comparable accuracy.

[Figure]

Figure R1: (a) Bar plots of snow depth measured manually and by GPS-IR using SNR within the 0–360° azimuth range. (b) Scatter plot of the manual snow depth and GPS-IR-estimated ones. The correlation coefficient (R), root-mean-square error (RMSE), and bias are presented. (c) and (d) are similar to (a) and (b), respectively, but for the GPS-IR measurements over the azimuth range of 90°–135°.

1.2 In addition, the way that the authors explained this "systematic" inconsistency does not make sense. The reflectivity difference between snow and the underlying frozen ground is not so much for GPS L-band signals. Moreover, this reflectivity difference, even if we consider it as a potential source of error, would not affect neither the amplitude nor the polarization because signals are assumed to be reflected off the "top" of the snowpack.

We compare the reflectivity between snow and the ground at the beginning of thawing season rather than the underlying frozen ground. In the framework of GPS-IR, the snow depth is derived as the difference between the reflector height of snow surface and the one of ground surface serving as reference (Fig. R2). In permafrost areas, the ground surface is subject to progressive subsidence in summer. If we use the mean value of the reflector heights during the

entire thaw season as reference (which is normally used in previous studies), a bias would be introduced into snow depth. Thus, in this study, we use the average reflector height at the beginning of the thawing season (i.e., DOY 167–173 in 2017) to be the reference (*Page 12, Lines 249–250*).

[Figure]

Figure R2: Schematic diagram showing the reflector heights in snow and snow-free conditions, denoted by $H_{snow}$ and $H_{ground}$. Snow depth is the difference between $H_{snow}$ and $H_{ground}$ serving as a reference. In this study, $H_{ground}$ refers to the average reflector height at the beginning of the thawing season.

At the beginning of thawing, the ground is not covered by snow and the soil starts to thaw downward from the surface. Based on the in situ measurements, the mean surface soil moisture at the depth of 1 cm during DOY 167–173 in 2017 was around 38% volumetrically. Given the significant difference of moisture content between snow and thawed soil, their difference in reflectivity for GPS signals cannot be ignored.

By only considering surface reflection, Zavorotny et al. (2010) developed a forward physical model to simulate reflected signals and SNR in a bare-soil condition. By varying soil moisture content, the top-soil reflectivity changes, then the amplitude and phase of the simulated SNR

vary correspondingly. Thus, even only considering surface reflection, the reflectivity of substrate has impact on SNR observations.

We use the open-source physical model of Nievinski and Larson (2014) to simulate SNR in the conditions of snow and wet soil, to investigate the impact of their reflectivity difference on reflector height retrieval. The key parameters for the simulations are presented in Table R1. The simulated SNR observations are presented in Figure R3(a). We can observe a clear difference between the amplitude of the SNR series. We conduct Lomb-Scargle Periodogram analysis on the SNR simulations to obtain their frequency spectrums, which are shown in Figure R3(b). The SNR in the snow case has a larger amplitude. The dominant reflector height corresponding to the peak power is 1.99 m, whereas 2.03 m for the wet soil. The difference in the reflectivity of snow and wet soil does affect SNR observations and introduce bias to reflector height retrievals consequently to snow depth measurements. In the simulations, the introduced bias is 4 cm, which makes the GPS-IR measurements overestimate the snow depth.

[Figure]

Figure R3: (a) Simulated SNR observations with the reflector of wet soil (black curve) and snow (blue curve). (b) Frequency spectrum of the simulated SNR observations. The frequency has been converted into reflector height. The dominant reflector height is 2.03 m for wet soil, whereas 1.99 m for snow.

Table R1: Key parameters used in the simulations

| Parameter | Value |
|---|---|
| Antenna (Radome) | TRM29659.00 (SCIT) |
| Signal | L1 C/A |
| Antenna height | 2 m |
| Azimuth range | 0–360° |
| Elevation angle | 5–20° |
| Reflector material #1 | Sandy loam with soil moisture content of 38% volumetrically |
| Reflector material #2 | Dry snow with default properties |

1.3 Furthermore, "possible penetration into the soil when manually probing the rod", if happened, would reduce the bias as it would cause an overestimation in in-situ snow depth measurements.

Thank you for the clarification. The penetration into soil when manually probing to measure snow depth is another possible error source. It would overestimate the snow depth and compensate to some extent the overestimation of the GPS-IR measurements. The bias of the GPS-IR measurements is the integrated impact from all of the error sources. Its sign depends on the magnitude of each impact factor. The reflectivity difference leads to positive, whereas the probing penetration results in negative. In the discussion of GPS-IR snow depth (*Page 17, Lines 322–330*), our objective is to clarify the error sources.

1.4 I would seek for either a better explanation or a reconsideration in the snow depth retrieval method. Looking into "higher-order frequencies" can be a solution for this issue as proposed by Cardellach, Fabra, et al. (2012) and Ghiasi (2020).

The "higher-order frequencies" means the other frequencies than the dominant first-order one with peak power in the frequency spectrum, if we understand this term correctly. Cardellach et al. (2012) used a dual-polarized (right and left-hand circular polarized, RHCP and LHCP in short)

side-looking antenna to receive reflections and developed a forward model to detect and identify the signals from internal layers a few hundred meters deep within the snow cover in Antarctica. In our study, the antenna is RHCP and in up-right direction. At low elevation angles, on one hand, the RHCP reflections are already weak due to de-polarization; on the other hand, the dominant reflection occurs at the air-snow interface, the basis of using GPS-IR to estimate snow depth. Thus, the reflections from the deeper layers are expected to be negligible and barely disturb the dominant reflections. Therefore, the dominant first-order frequency should be used for estimating snow depth in our study.

As for Ghiasi et al. (2020, Application of GNSS interferometric reflectometry for the estimate of lake ice thickness), they put an antenna directly on the lake ice to receive the reflected signals off the interface between ice and underneath water. They also utilized the dominant frequency of SNR observations to obtain the distance between antenna and ice-water interface then lake ice thickness.

Given the better agreement of our new GPS-IR-estimated snow depth (Fig. R1), we believe it is unnecessary to use other methods to estimate snow depth at our site.

2. The authors have used Stefan's equation for modeling the surface elevation changes as they believe GPS-IR elevation retrievals are not accurate enough because their uncertainties are in the order of few centimetres. I would say that "a few centimetres" is an acceptable accuracy for this purpose since Stefan's equation has not shown a better accuracy in literature. I would suggest conduct the same validation using surface elevations directly obtained by GPS-IR. Besides, the term "uncertainty" used by the authors does not look very exact because it is driven based on the standard deviation of the mean values which are not necessarily to be normally distributed.

We separate this comment into two points and address them correspondingly.

2.1 The authors have used Stefan's equation for modeling the surface elevation changes as they believe GPS-IR elevation retrievals are not accurate enough because their uncertainties are in

the order of few centimetres. I would say that "a few centimetres" is an acceptable accuracy for this purpose since Stefan's equation has not shown a better accuracy in literature. I would suggest conduct the same validation using surface elevations directly obtained by GPS-IR.

The reason we do not use the GPS-IR-estimated surface elevation changes directly is that they have relatively large daily oscillations. In reality, the ground surface typically subsides progressively. The GPS-IR measurements cannot represent the real evolution of surface subsidence on a daily scale. Alternatively, we turn to use the Stefan model to fit the GPS-IR measurements to obtain the smoothed seasonal time series. We have revised the manuscript to remove the misleading contents.

In Fig. R4, we show the comparison between the in situ measurements and the GPS-IR ones by using the default method, our improved method, and the GPS-IR surface elevation changes directly (denoted by raw data for simplicity). We can observe that the GPS-IR results by our improved method are more reliable compared with the ones by default method and the raw data directly. The raw-data method has the worst performance, giving a moderate-to-low correlation coefficient of 0.43 and an RMSE of 2.34%.

[Figure]

Figure R4: Comparison between in situ soil moisture and GPS-IR measurements estimated by using default method, our improved method, and GPS-IR-estimated surface elevation changes (denoted by raw data for simplicity) directly. The correlation coefficient (R), bias, and root-mean-square error (RMSE) are presented as well. We do not present the uncertainties of the GPS-IR estimates for clarity.

2.2 Besides, the term "uncertainty" used by the authors does not look very exact because it is driven based on the standard deviation of the mean values which are not necessarily to be normally distributed.

On any given day, multiple SNR interferograms are available. We retrieve the reflector heights from these SNR data initially, then use their mean to be the daily measurement. The uncertainty is the standard deviation of the mean value. The following figure shows the

distribution of reflector heights on four given days, i.e., DOY 190, 200, 210, and 220 in 2018, and their normal distribution fit. We can find that the distribution of the reflector height on each given day generally follows the normal distribution.

Following the reviewer's suggestion, we have replaced the uncertainty with the standard deviation of the mean.

[Figure]

Figure R5: Histograms of reflector heights on the given days of DOY 190, 200, 210, and 220 in 2018, and their normal distribution fit. The mean ($\mu$) and standard deviation ($\sigma$) are also presented (units: m).

3. Although the paper appears in a very clear and accurate English writing, some sentences are too short, e.g., line 221, and some sentences start with "And" which looks somehow inappropriate in academic English writing, e.g., lines 236 and 266.

We have revised the manuscript accordingly.

References:

Cardellach, E., Fabra, F., Rius, A., Pettinato, S. and D'Addio, S.: Characterization of dry-snow sub-structure using GNSS reflected signals, Remote Sens. Environ., 124, 122–134, doi:10.1016/j.rse.2012.05.012, 2012.

Ghiasi, Y., Duguay, C. R., Murfitt, J., van der Sanden, J. J., Thompson, A., Drouin, H. and Prévost, C.: Application of GNSS Interferometric Reflectometry for the Estimation of Lake Ice Thickness, Remote Sens., 12(17), 2721, doi:10.3390/rs12172721, 2020.

Larson, K. M.: Unanticipated Uses of the Global Positioning System, Annu. Rev. Earth Planet. Sci., 47(1), 19–40, doi:10.1146/annurev-earth-053018-060203, 2019.

Nievinski, F. G. and Larson, K. M.: An open source GPS multipath simulator in Matlab/Octave, GPS Solut., 18(3), 473–481, doi:10.1007/s10291-014-0370-z, 2014.

Zavorotny, V. U., Larson, K. M., Braun, J. J., Small, E. E., Gutmann, E. D. and Bilich, A. L.: A Physical Model for GPS Multipath Caused by Land Reflections: Toward Bare Soil Moisture Retrievals, IEEE J. Sel. Top. Appl. Earth Obs. Remote Sens., 3(1), 100–110, doi:10.1109/JSTARS.2009.2033608, 2010.